# Brain Delivery of IGF1R5, a Single-Domain Antibody Targeting Insulin-like Growth Factor-1 Receptor

**DOI:** 10.3390/pharmaceutics14071452

**Published:** 2022-07-12

**Authors:** Alvaro Yogi, Greg Hussack, Henk van Faassen, Arsalan S. Haqqani, Christie E. Delaney, Eric Brunette, Jagdeep K. Sandhu, Melissa Hewitt, Traian Sulea, Kristin Kemmerich, Danica B. Stanimirovic

**Affiliations:** Human Health Therapeutics Research Centre, National Research Council Canada, Ottawa, ON K1A 0R6, Canada; alvaro.yogi@nrc-cnrc.gc.ca (A.Y.); greg.hussack@nrc-cnrc.gc.ca (G.H.); henk.vanfaassen@nrc-cnrc.gc.ca (H.v.F.); arsalan.haqqani@nrc-cnrc.gc.ca (A.S.H.); christie.delaney@nrc-cnrc.gc.ca (C.E.D.); eric.brunette@nrc-cnrc.gc.ca (E.B.); jagdeep.sandhu@nrc-cnrc.gc.ca (J.K.S.); melissa.hewitt@nrc-cnrc.gc.ca (M.H.); traian.sulea@cnrc-nrc.gc.ca (T.S.); kristin.kemmerich@nrc-cnrc.gc.ca (K.K.)

**Keywords:** blood–brain barrier, receptor-mediated transcytosis, single domains antibody, IGF1R, neurotensin

## Abstract

The ability of drugs and therapeutic antibodies to reach central nervous system (CNS) targets is greatly diminished by the blood–brain barrier (BBB). Receptor-mediated transcytosis (RMT), which is responsible for the transport of natural protein ligands across the BBB, was identified as a way to increase drug delivery to the brain. In this study, we characterized IGF1R5, which is a single-domain antibody (sdAb) that binds to insulin-like growth factor-1 receptor (IGF1R) at the BBB, as a ligand that triggers RMT and could deliver cargo molecules that otherwise do not cross the BBB. Surface plasmon resonance binding analyses demonstrated the species cross-reactivity of IGF1R5 toward IGF1R from multiple species. To overcome the short serum half-life of sdAbs, we fused IGF1R5 to the human (hFc) or mouse Fc domain (mFc). IGF1R5 in both N- and C-terminal mFc fusion showed enhanced transmigration across a rat BBB model (SV-ARBEC) in vitro. Increased levels of hFc-IGF1R5 in the cerebrospinal fluid and vessel-depleted brain parenchyma fractions further confirmed the ability of IGF1R5 to cross the BBB in vivo. We next tested whether this carrier was able to ferry a pharmacologically active payload across the BBB by measuring the hypothermic and analgesic properties of neurotensin and galanin, respectively. The fusion of IGF1R5-hFc to neurotensin induced a dose-dependent reduction in the core temperature. The reversal of hyperalgesia by galanin that was chemically linked to IGF1R5-mFc was demonstrated using the Hargreaves model of inflammatory pain. Taken together, our results provided a proof of concept that appropriate antibodies, such as IGF1R5 against IGF1R, are suitable as RMT carriers for the delivery of therapeutic cargos for CNS applications.

## 1. Introduction

The development of therapies for central nervous system (CNS) indications is hampered by several factors, including poor delivery due to the blood–brain barrier (BBB). Tight junctions between the endothelial cells forming the BBB prevent the paracellular transport of most synthetic drugs and large molecules, such as biologics [1]. The brain delivery of essential macromolecules and nutrients can be achieved via receptor-mediated transcytosis (RMT)-dependent and RMT-independent mechanisms [2,3,4]. RMT is initiated by ligand binding to a receptor on the luminal surface of brain endothelial cells (BECs). The ligand–receptor complex undergoes trafficking through multiple intracellular endosomal compartments where the cargo is detached from the receptor and then released on the abluminal side of the barrier. Meanwhile, the receptor recycles ‘back’ to accept additional cargo molecules. Targeting this endogenous mechanism of transcytosis is an attractive approach to delivering therapeutic cargos, especially macromolecules, across the BBB [5,6,7].

Currently, the main RMT receptors that have been studied are the transferrin receptor (TfR) and insulin receptor (IR), and ligands against these receptors were shown to deliver different therapeutic cargos into the brain [8,9,10]. Additional targets shown to mediate RMT include insulin-like growth factor receptors (IGF1R) and transmembrane protein 30A (TMEM30A/CDC50A). It should be noted that several other targets, including low-density lipoprotein receptor (LDLR), low-density lipoprotein-related protein 1 (LRP-1), CD98hc, LRP8 and others, were implicated in BBB transcytosis, although the exact mechanisms of their BBB crossing remain unclear [11,12,13,14,15,16,17,18,19].

We previously developed camelid single-domain antibodies (sdAbs, V_H_Hs) against some of these target receptors (TMEM30A/CDC50A, IGF1R) and demonstrated the feasibility of antibody-mediated drug delivery via the RMT pathway [11,12,13]. In addition, it was shown that drug cargos can be incorporated into liposomes or nanoparticles decorated with the RMT-targeting ligand to boost brain delivery [17,20]. However, when compared with conventional antibodies and nanotechnologies, camelid sdAbs present numerous advantages for this application, including their small size, ease of engineering, optimization and humanization, strong biophysical properties and low immunogenicity.

Insulin-like growth factor-1 receptor (IGF1R) was identified as a potential RMT candidate based on the observation that its ligand IGF-1 was transported across the BBB and its elevated expression in BECs relative to peripheral tissue [21]. SdAbs targeting the ectodomain of IGF1R were isolated via llama immunization, and their transmigration was demonstrated in rat and human BBB models in vitro [22,23]. We further confirmed these findings in vivo by showing that three of the sdAbs isolated from the initial panning displayed increased accumulation in the brains of rats and mice [11]. By isolating brain microvessel and parenchymal fractions followed by mass spectroscopy quantification of antibodies, we were able to quantify the IGF1R4 sdAb that was shuttled into the brain parenchyma versus the fraction bound or accumulated inside the endothelial cells [11].

One of the potential side effects associated with RMT targets is interfering with their normal physiological functions. To mitigate this possibility, we recently mapped the binding epitope of one of the BBB-crossing sdAbs, namely, IGF1R5, on IGF1R in relation to IGF-1 using differential hydrogen–deuterium exchange mass spectrometry and nuclear magnetic resonance spectroscopy [24]. Furthermore, we demonstrated that this IGF1R sdAb has no detectable impact on the functional activation of IGF1R. Whether this sdAb variant is able to effectively deliver a pharmacologically active payload across the BBB remains to be determined. The ability of mFc and hFc fusions in variable C- or N-terminus linkages of IGF1R5 to cross the BBB, as well as their ability to shuttle a pharmacologically active payload across the BBB, was confirmed in this study by analyzing the hypothermic properties of neurotensin when fused to IGF1R5hFc constructs. Furthermore, we demonstrated that IGF1R5 humanization by modifications in the backbone structure of IGF1R5 did not affect its BBB permeability. The present study provided a proof of concept and validated IGF1R5 as an RMT receptor ligand that is suitable for the delivery of therapeutic cargos for CNS applications.

## 2. Materials and Methods

### 2.1. V_H_H Isolation

Llama single-domain antibodies (V_H_Hs) against IGF1R were isolated and produced as described previously [23]. Briefly, one male llama (*Lama glama*) was immunized with the extracellular domain of human IGF1R consisting of 933 amino acids. The antigen-specific immune response was monitored at different time points post-immunization, and on day 84, peripheral blood mononuclear cells (PBMCs) were collected for sdAb phage display library construction and panning. IGF1R5 sdAb isolated through library panning were subcloned and expressed in TG1 Escherichia coli cells and purified using HiTrap Chelating HP columns (GE Healthcare, North Richland Hills, TX, USA).

### 2.2. Humanization

The camelid IGF1R5 sdAb sequence was humanized following a CDR grafting protocol [25]. Briefly, a human VH3 germline was used as a framework (FR) template. The complementarity-determining region (CDR) was defined according to the Kabat definition and sequence numbering. Back-mutations were selected to arrive at several humanized variants based on multiple criteria, among which proximity to CDR required 3D structural homology modeling of the camelid V_H_H. In the case of the H2 humanized variant, camelizing back-mutations in the FR2 were not introduced. Instead, only 4 back-mutations were considered for this variant, all of which were in the Vernier zone supporting the CDR loops. Additionally, the CDR2 point mutation A57T was introduced to enable scalable purification using Protein A affinity chromatography [26] if required for future large-scale biomanufacturing. The mutations introduced in the humanized IGF1R5-H2 relative to the camelid IGF1R5 sdAbs are highlighted in Appendix A. The humanized IGF1R5-H2 sdAb was produced and purified as described for the llama sdAbs.

### 2.3. Surface Plasmon Resonance (SPR) Binding of IGF1R5 and IGF1R5-H2 V_H_Hs to IGF1R

The affinities of wild-type IGF1R5 and humanized IGF1R5-H2 V_H_Hs for several IGF1R ectodomain orthologues (human, rhesus, mouse and rat) were determined using SPR. Immediately prior to SPR, V_H_Hs were purified using preparative size exclusion chromatography (SEC) to isolate pure monomeric V_H_H fractions. SEC was performed by injecting 250–300 µg of each V_H_H over a Superdex S75 Increase 10/300 GL column (Cytiva, Marlborough, MA, USA) controlled by an ÄKTA FPLC Purifier (Cytiva) at a flow rate of 0.8 mL/min in HBS-EP buffer (10 mM HEPES, pH 7.4, containing 150 mM NaCl, 3 mM EDTA and 0.005% *v/v* surfactant P20 (polyoxyethylene 20 sorbitan monolaurate); Cytiva). All SPR experiments were performed on a Biacore 3000 and a Biacore T200 (Cytiva) at 25 °C in an HBS-EP buffer. Ectodomains of human IGF1R (R&D Systems, Cat#391-GR-050), rhesus IGF1R (NRC Montreal, aa31-932), rat IGF1R (NRC Montreal, aa31-936) and mouse IGF1R (R&D Systems, Cat#6630-GR/CF-025) were amine coupled on CM5 sensor chips (Cytiva) at 10 µg/mL in 10 mM acetate pH 4.0 using an amine-coupling kit (Cytiva), resulting in approximately 1500–2000 response units (RUs) of each IGF1R ectodomain immobilized. The remaining active sites were blocked with 1 M ethanolamine at pH 8.5. An ethanolamine-blocked empty flow cell served as a reference surface. On the Biacore 3000, using multi-cycle kinetics (MCK), V_H_Hs at various concentration ranges were injected over the IGF1R ectodomains and reference surface at a flow rate of 20 µL/min for 300 s followed by 300 s of dissociation. On the Biacore T200, using single-cycle kinetics (SCK), V_H_Hs were injected at 40 µL/min for 180 s followed by 600 sec of dissociation. The V_H_H concentration ranges were 0.25–10 nM (IGF1R5) and 1–25 nM (IGF1R5-H2). Surfaces were regenerated with a 24 s pulse of 10 mM glycine, pH 2.0, at a flow rate of 100 µL/min. Reference subtracted sensorgrams were analyzed and fit to a 1:1 binding model with BIAevaluation 4.1 software (Biacore 3000) or BIAevaluation 3.2 software (Biacore T200; Cytiva, Marlborough, MA, USA). IGF1R5 and IGF1R5-H2 affinities for human and mouse IGF1R were also determined at pH 5.6 using an HBS-EP MES buffer (10 mM HEPES, 10 mM MES, 150 mM NaCl, 3 mM EDTA, 0.005% P20, pH 5.6) at 37 °C. Prior to injection, V_H_Hs were buffer exchanged using Amicon ultra-centrifugal filters (0.5 mL, 3K MWCO). IGF1R5 flowed at 0.25–10 nM and IGF1R5-H2 flowed at 1–50 nM with similar contact times, dissociation times, regeneration conditions and fitting as those described above.

### 2.4. Rat and Human BBB Models In Vitro

Simian virus 40-immortalized adult rat brain endothelial cells (SV-ARBECs) were seeded at 80,000 cells/membrane on rat-tail-collagen-coated 0.83 cm^2^ Falcon cell inserts (1 μm pore size) in 1 mL SV-ARBEC culture medium without phenol red. The inserts were placed in the wells of a 12-well tissue culture plate containing 1 mL of SV-ARBEC medium without phenol red and 1 mL rat astrocyte-conditioned medium to generate an in vitro model of the BBB as described previously [11,12,27]. Permeability was monitored and the cultures were used only when Pe [sucrose] was between 0.4 and 0.6 (×10^−3^) cm/min. Transport experiments were performed by adding an equimolar mixture of antibodies to the top chamber and collecting a 100 μL aliquot from the bottom chamber at 15, 30, 60 and 90 min for simultaneous quantification of the antibodies using the SRM method. Control antibodies of the same were added to each transport well to determine the background transport resulting from paracellular/nonspecific flux. The apparent permeability coefficient Papp was calculated using Papp = ΔQ/Δt × 1/AC_0_, where ΔQ/Δt is the steady-state flux (mol/min), A is the surface area of the filter (cm^2^) and C_0_ is the initial concentration in the top chamber.

Human-amniotic-fluid-derived induced pluripotent stem cells (AF-iPSCs) were generated from human amniotic fluid (AF) cells and differentiated into iBECs as previously described [22]. Briefly, AF-iPSCs were seeded at a density of 8  ×  10^3^ cells/cm^2^ in DMEM/F12 medium (Thermo Fisher Scientific, Waltham, MA, USA) supplemented with 20% KnockOut Serum Replacement, 1  ×  Glutamax, 1  ×  Non-Essential Amino Acids and 0.1 mM β-mercaptoethanol (all from Life Technologies, Carlsbad, CA, USA) for 6 days. The medium was changed to EM medium (human Endothelial Serum-Free medium, Life Technologies) supplemented with 20 ng/mL basic fibroblast growth factor (bFGF, Life Technologies), 10 μM retinoic acid (RA, Sigma Aldrich, Oakville, ON, Canada) and 1% fetal bovine serum (FBS, Hyclone Laboratories, Logan, UT, USA) for an additional 2 days. To establish the in vitro transwell BBB model, iBECs were dissociated with Accutase (Stem Cell Technologies, Vancouver, BC, Canada) and seeded at a density of 2.5 × 10^5^ cells per 24 well transwell insert (3 µm pore size, 0.33 cm2 surface area; BD, Mississauga, ON, Canada) pre-coated with collagen type-IV (80 µg/mL, Sigma) and fibronectin (20 µg/mL, Sigma) in complete EM medium with 10 µM Y27362 (ROCK Inhibitor, Stem Cell Technologies). iBEC transwells were incubated overnight at 37 °C in 5% CO_2_ and the next day, the medium was changed to an EM medium without bFGF and RA for an additional 24 h in the luminal chamber. Antibody transport experiments and apparent permeability coefficient calculations were performed as described above.

### 2.5. NanoLC-SRM Mass Spectrometry Analyses

The nanoflow ultrahigh performance liquid chromatography-coupled selected-reaction monitoring (nanoLC-SRM) mass spectrometry method was used to quantify absolute or relative levels of proteins in a BBB model, serum, cerebrospinal fluid and vessel-depleted brain parenchyma fractions. All protein extracts were reduced, alkylated and trypsin-digested using a previously described protocol [28]. Mass spectrometry analyses were carried out on a NanoAcquity UPLC (Waters, Milford, MA, USA) containing a C18 PepMap™ 100 trap (ThermoFisher, Waltham, MA, USA) followed by a nanoLC BEH130C18 column (Waters) coupled with ESI-LTQ-XL-ETD or ESI-TSQ-Quantiva mass spectrometers (ThermoFisher). Peptide signatures of various antibodies and vessel/parenchymal markers were identified by analyzing the respective samples with tandem mass spectrometry (nanoLC-MS/MS) using data-dependent acquisition on ESI-LTQ-XL-ETD44. For the absolute quantification of antibodies, at least 9 standards consisting of calibration and QC standards between 0.05 and 16 fmol range were created by spiking antibodies in their respective control matrices. Each sample was analyzed using nanoLC-SRM and data was extracted from raw files and analyzed using Skyline 64-bit 20.2.0.286 software (MacCoss Lab Software, University of Washington, Seattle, WA, USA) available as open source software from https://skyline.ms (last accessed on 11 July 2022).

### 2.6. Animals

Male Wistar rats (weight range, 150–200 g) and CD-1 mice (22–30 g) were purchased from Charles River Laboratories, Inc. (Montreal, QC, Canada). Animals were housed in groups of 3 in a 12 h light–dark cycle at a temperature of 24 °C, relative humidity of 50 ± 5% and were allowed free access to food and water. All animal procedures were approved by the NRC’s Animal Care Committee and were in compliance with the Canadian Council of Animal Care guidelines.

### 2.7. Serum, Cerebrospinal Fluid and Brain Exposure

All compounds were administered to rats via the tail vein. Twenty-four hours post-injection, rats were anesthetized with 3% isoflurane and CSF was collected from a direct puncture to the cisterna magna. Blood samples were taken from the tail vein following CSF collection and samples were centrifuged (15 min at 15,000 rpm; room temperature). Samples were stored at −80 °C until analysis.

Following blood collection, rats were thoroughly perfused with 10 mL of heparinized (100 U/mL) saline at a rate of 1 mL/min via the left common carotid artery to facilitate specific perfusion of the brain. Brains were then removed and homogenized in an ice-cold homogenization buffer containing 50 mM Tris-HCl pH 8, 150 mM NaCl, and a protease inhibitor cocktail (Sigma-Aldrich, Oakville, ON, Canada) using a Dounce homogenizer (10–12 strokes at 4 °C). Brain homogenates were depleted of vessels using a sequential filtration through 100 and 20 μm nylon Nitex mesh filters (pluriSelect, Leipzig, Germany). Successful vascular depletion of parenchymal fractions was confirmed using the enrichment of a parenchymal marker (Slc1a3) with the concomitant absence of a specific vascular marker (Slc2a1) as previously observed [11]. The concentration of injected antibodies was determined in vessel-depleted parenchymal fraction using SRM as described above.

### 2.8. Immunofluorescence

Brains were removed from the skull and drop fixed in 4% paraformaldehyde for 24 h at room temperature, followed by cryoprotection in 30% sucrose solution for 48 h at 4 °C. Brains were then embedded in an Optimal Cutting Temperature compound (OCT), frozen over dry ice and stored at −80 °C until sectioning. Coronal sections were cut at 15 µm, mounted on Superfrost plus slides (Thermo Fisher Scientific, Waltham, MA, USA) and subjected to immunofluorescent staining. Sections were incubated with DAKO serum-free protein block (DAKO Diagnostics, Burlington, Canada) containing 0.25% Tween-20 for 40 min at room temperature, followed by overnight incubation at 4 °C. The following antibodies were used: goat anti-mouse-IgG Fcγ-cy3 (1:200, Cat#115-165-071, Jackson ImmunoReasearch, West Grove, PA, USA), mouse-anti-NueN (1:100, Cat# ab13938, Abcam, UK) and RCAI (1:500, Cat# FL-1081, Vector Laboratories, Newark, NJ, USA). Following overnight incubation, sections were washed 3× with Tris-buffered saline (TBS, DAKO) and the conjugate was detected via incubation for 45 min at room temperature with 1:300 goat anti-mouse Alexa 647 (A21235, Invitrogen, Waltham, MA, USA) or 1:500 goat anti-rabbit Alexa 647 (A21244, Invitrogen, Waltham, MA, USA). After washing 3× with TBS, sections were mounted in DAKO fluorescent mounting medium and spiked with Hoechst (2 µg/mL, Cat#H3570, Invitrogen, Waltham, MA, USA). Images were captured with an Olympus 1 × 81 Fluorescent Microscope using 10× and 60× objectives and following the manufacturer’s instructions for excitation and emission channels.

### 2.9. Expression and Purification of IGF1R5 Fusions with Fc Domain and Neurotensin

DNA encoding hFc-IGF1R5, IGF1R5-mFc, mFc-IGF1R5 and IGF1R5-hFc-neurotensin was synthesized using Genescript. A schematic showing different constructs/fusion proteins used in this study is presented in Figure 1. The sequences for IGF1R5, mouse IgG Fc2b and human IgG Fc fragment were as previously described [12,23]. The IGF1R5-hFc-neurotensin sequence includes a linker (amino acid sequence GGGSGGGGS). Constructs were expressed in transiently transfected Chinese hamster ovary cells (CHO-3E7). The culture medium was harvested 7 days post-transfection via centrifugation and clarified using 0.2 µm filter bottles (Millipore Stericup, MilliporeSigma, Burlington, VT, USA). Clarified medium was applied on a column packed with 5 mL (volume of columns used depended on protein titer and volume of culture) protein-A MabSelect SuRe resin (GE Healthcare, Mississauda, ON, Canada). After loading, the column was washed with 5–10 volumes of phosphate-buffered saline pH 7.1 (PBS) and the constructs were eluted with 100 mM sodium citrate buffer pH 3.0 to 3.6. Then, a buffer exchange was performed by loading on a desalting NAP-25 column (GE Healthcare, Mississauda, ON, Canada) equilibrated in PBS. Desalted constructs were then sterile-filtered by passing through a Millex GP (MilliporeSigma, Burlington, VT, USA) filter unit (0.22 µm) and aliquoted. The purity of the protein was verified using SDS-PAGE and they were stored at −80 °C.

### 2.10. Cell-Based Neurotensin Receptor 1 Activation

A receptor functional assay was carried out using the PathHunter eXpress NTR1 kit (DiscoverX, Fremont, CA, USA). Briefly, engineered HEK293 cells expressing a (Pro-Link or PK)-tagged NTR1 and an enzyme acceptor (EA)-tagged SH2 domain were used. Upon receptor activation, EA-SH2 binds to the phosphorylated NTR1 and reconstitutes an active β-galactosidase enzyme, which hydrolyzes the substrate to generate a chemiluminescent readout. Cells were thawed and plated in a 384-well white-wall clear-bottom plate (Greiner, Monroe, LA, USA) at 20,000 cells/well for 24 h at 37 °C in 5% CO_2_ following the manufacturer’s instructions. Cells were then treated with neurotensin (Sigma-Aldrich, Oakville, ON, Canada) or IGF1R5-hFc-neurotensin. The chemiluminescent substrate was added and cells were incubated at RT for 60 min. The resulting luminescence was measured using the CLARIOstar plate reader (BMG LABTECH, Ortenberg, Germany) and concentration–response curves were generated using nonlinear regression analysis (GraphPad Prism Software, San Diego, CA, USA).

### 2.11. IGF1R5-hFc-Neurotensin-Induced Hypothermia in Rats and Mice

Wistar rats and CD-1 mice were used. Before surgery, animals were injected with sustained-release buprenorphine (1.2 mg/kg) subcutaneously for analgesia. Temperature data loggers were implanted in the peritoneal cavity of rats (DST micro-T, Star-Oddi, Gardabaer, Iceland) and mice (DST nano-T, Star-Oddi, Gardabaer, Iceland) under isoflurane anesthesia. Animals were allowed to recover from surgery for 1 week prior to the injection of the test compound.

Data recording on temperature loggers was initiated 48 h prior to injection for calculations of baseline values. Intravenous injection of the test compound was performed between 7:00 AM and 9:00 AM by experienced personnel to avoid stress-induced hyperthermia. The data loggers measured the core body temperature of animals at a time interval of 1 min to an accuracy of 0.1 °C for up to 6 h post-injection.

Core temperature baseline values were taken in undisturbed animals 24 h or 48 h prior to the test compound injection. To avoid variability due to regular changes in temperature during the circadian cycle, the start point and time frame of baseline values matched that of test compound injection. The average baseline value (Tb) and standard deviation (SD) of baseline values were used to calculate the duration of response. Hypothermia duration was defined as the time in which Tc < Tb–2SD during the interval from dosing up to 240 min in rats and 360 min in mice. The maximum change in core body temperature was expressed as the difference measured at the minimum temperature observed after the test compound injection compared with the baseline core body temperature measured as described above. The area under the curve (AUC) was calculated using the trapezoidal rule from 0 to 240 min and 360 min in rats and mice, respectively. The AUC calculation was performed using GraphPad Prism.

### 2.12. IGF1R5mFc-Galanin and mFcIGF1R5-Galanin-Induced Analgesia in Rats

To further demonstrate whether IGF1R5 constructs in C- and N-terminal fusion to mFc can cross the BBB in vivo and deliver a molecule that cannot cross the BBB on its own, the neuropeptide galanin was chemically conjugated to IGF1R5 and administered systemically as previously described [29]. Galanin is a neuroactive peptide that produces analgesia by binding GalR1 and GalR2 expressed in brain tissue. When given peripherally, galanin has no analgesic effects because it cannot cross the BBB on its own.

IGF1R5-mFc and mFc-IGF1R5 were conjugated to a rat galanin fragment with a cysteamide-modified C-terminus (Biomatik, Cambridge, ON, Canada) (GWTLNSAGYLLGPHAIDNHRSFSDKHGLT-cysteamide) as previously described [29]. Briefly, 2 mg of each IGF1R5 was placed in 4 separate 1.5 mL micro-centrifuge tubes and diluted to 2 mg/mL with PBS. Sulfo-SMCC was added in a 6.5x excess molar ratio; specifically, 29.5 μL of the 2.5 mg/mL Sulfo-SMCC was added to each micro-centrifuge tube. The micro-centrifuge tubes containing the mixture were incubated for 30 min at room temperature (RT) with short vortexing every 10 min. Once the reaction was done, the unreacted Sulfo-SMCC was removed from the maleimide-activated IGF1R5 using a 10 mL 7K Zeba column (Pierce). Prior to sample loading, the column was washed 3 times with 5 mL PBS and spun at 1000× *g* for 2 min. The 4 separate reactions were combined and loaded on the column. The column was spun for 2 min at 1000× *g*.

Separately and concurrently, a 1 mg/mL stock of galanin-cysteamide was prepared in Milli-Q H_2_O. The purified maleimide-activated IGF1R5 constructs were mixed with galanin-cysteamide, sealed and incubated overnight at 4 °C or 1 h at RT. The unreacted galanin-cysteamide was removed using Amicon-15 30K column (MilliporeSigma, Oakville, ON, Canada). The samples were added to the column, and the volume was filled to 15 mL with PBS and spun at 4000× *g* for 7 min until the volume was reduced to 2 mL. The conjugated sample was then added to a 5 mL 7K Zeba column (ThermoFisher Scientific, Waltham, MA, USA) prepared as described above (wash was done with 2.5 mL PBS), and then spun for 2 min at 1000× *g*. The collected sample comprised the IGF1R5-mFc-galanin and mFc-IGF1R5-galanin conjugation product. The protein concentration was determined by measuring the absorbance at A280 on a NanoDrop. The reaction was titrated to achieve about 1 to 2 galanin molecules per construct. A reaction was confirmed by loading and silver staining conjugated IGF1R5-mFc-Gal and mFc-IGF1R5-Gal samples on a 10% SDS-PAGE to confirm a shift in molecular weight size after conjugation.

The Hargreaves model of hyperalgesia was used to evaluate the efficacy of IGF1R5 to deliver galanin into the brain and induce antinociceptive effects, as previously demonstrated [11,12,29]. Chronic inflammatory hyperalgesia was induced in one of the paws of Wistar rats by injecting 100 μL of complete Freund’s adjuvant (CFA; heat-killed Mycobacterium tuberculosis; Sigma–Aldrich, Oakville, ON, Canada) suspended in an oil:saline (1:1) emulsion. Then, the plantar surface of both the right and left paw was exposed to a radiant stimulus and the paw withdrawal latency (PWL) of each paw was measured using the plantar Analgesia Meter equipment for paw stimulation (IITC Life Science, Woodland Hills, CA, USA). The time spent between starting the radiant exposure and clicking or flicking the paw was interpreted as a positive nociceptive response. A cut-off of 20 s was established to avoid tissue damage. Three days post-CFA injection, inflammatory hyperalgesia was confirmed by measuring the baseline PWL of the right and left paws. Test compounds were then administered intravenously through the tail vein and a time-course of the antinociceptive response was determined. The experimenter performing pain experiments was blinded to the contents of the injectable compounds. The area under the curve (AUC) was calculated by the trapezoidal method to derive the percentage of maximal possible effect (%MPE) using the formula %MPE = [(AUCmolecule-AUCinflamed paw)/(AUCnormal paw-AUCinflamed paw)] × 100, where AUCinflamed paw and AUCnormal paw are the values obtained from the group injected with the vehicle (PBS).

### 2.13. Statistical Analysis

The results are expressed as the mean ± SEM or SD as indicated. Where applicable, a paired *t*-test was used. One-way ANOVA followed by Newman-Keuls’ post-test was used to compare multiple groups. A *p*-value of less than 0.05 was considered statistically significant.

## 3. Results

### 3.1. Surface Plasmon Resonance (SPR) Binding of IGF1R5 and IGF1R5-H2 V_H_Hs to IGF1R

SPR was used to determine the affinities of the V_H_Hs for various IGF1R ectodomain orthologues (Table 1, Figure 2). IGF1R5 bound to IGF1R from all species tested, with affinities of KD = 0.6 nM, 0.4 nM, 1.1 nM and 1.1 nM for human, rhesus, mouse and rat IGF1R, respectively. To avoid potential immunogenicity of llama-derived IGF1R5 when applied as a BBB carrier for therapeutics, this V_H_H was humanized using mutations of defined ‘camelid’ residues in the parental molecule. In the case of the IGF1R5-H2 variant, four back-mutations in the Vernier zone supporting the CDR loops were introduced. This humanized version showed similar cross-reactivity, albeit with slightly weaker binding affinities of *K*_D_ = 7.6 nM, 17 nM, 9.1 nM and 11 nM for human, rhesus, mouse and rat IGF1R. V_H_H affinities for human and mouse IGF1R were also determined at pH 5.6 (37 °C). IGF1R5 was bound with comparable affinities to that observed at neutral pH, whereas IGF1R5-H2 possessed weaker binding affinity at acidic pH (*K*_D_ = 150–160 nM), with a faster off-rate (*k*_d_), which is a desirable characteristic for BBB crossing since the low pH present in endosomal trafficking would facilitate the sorting process, releasing the carrier from its receptor and sorting the membrane proteins and the carrier into different domains.

### 3.2. In Vitro and In Vivo BBB Transport of IGF1R5

The BBB permeability of IGF1R5 in the V_H_H format, the humanized H2 variant of IGF1R5 and the hFc-IGF1R5H2, along with their respective negative controls, were assessed in rat (SV-ARBEC) and human (iBEC) BBB models in vitro. Negative controls were A20.1 sdAbs generated against *C. difficile* toxin A, which has no recognizable mammalian target that does not cross the intact BBB. Figure 3A demonstrates that all formats were transported across both BBB models. Interestingly, whereas the humanized variant of IGF1R5 displayed increased Papp values in the rat model, its Papp value was similar to the other constructs in the human model. Next, the rat BBB model in vitro was used to determine the rate of transport of IGF1R5 and a negative control, namely, A20.1 with either C- or N-terminus mFc fusion. The molecules were co-applied to the upper chamber and quantified using targeted proteomics (SRM) in the bottom compartment. The Papp value for each antibody was calculated over 90 min. Figure 3B demonstrates that IGF1R5 constructs fused with mFc in both C- and N-terminus displayed enhanced transcytosis when compared with their counterpart negative controls. Although slightly higher, the Papp value for the IGF1R5–N-terminus mFc fusion (mFc-IGF1R5) was not significantly different from the IGF1R5–C-terminus mFc fusion (IGF1R-mFc). This is an important property of IGF1R5 not typically observed with other sdAbs since it allows for a ‘platform’ use of IGF1R5 in different types of fusion proteins where the specific orientation of the therapeutic cargo is important for its activity, most notably, therapeutic antibodies, where the addition of the BBB carrier to the C-terminus allows for the unimpeded activity of its target-binding Fab portion.

We next investigated the ability of IGF1R5 constructs to transport across the BBB in vivo by injecting hFc-IGF1R5 intravenously into rats (15 mg/kg) and measuring their CSF and capillary-depleted brain parenchyma levels 24 h post-injection. Figure 3C indicates that no difference in serum concentration was observed between hFc-IGF1R5 and the negative control used, namely, A20.1-mFc. On the other hand, both the CSF and brain levels of hFc-IGF1R5 were significantly increased when compared with A20.1-mFc, further highlighting the capacity of IGF1R5 constructs to be transported across the BBB. We also performed immunofluorescence staining to determine the localization of IGF1R5-mFc and A20.1-mFc in brain tissue following a single intravenous injection (15 mg/kg). Whereas a co-localization of IGF1R-mFc with RCA-1 (Figure 3E) suggested the presence in brain vasculature, there was also some co-localization with neurons (NeuN stain, Figure 3F), further supporting our observation that IGF1R5-mFc was able to cross the BBB and reach the brain parenchyma. As expected, A20.1-mFc staining was not observed in either brain vasculature (or was occasionally seen due to perfusion artifact) or in neurons (Figure 3G,H).

In addition to hFc fusions, we further increased the molecular weight of the constructs by generating and injecting animals with Ig fusions of IGF1R5 (Appendix A). We observed that the IgG-IGF1R5 concentrations in CSF and brain but not serum were significantly increased when compared with Ig alone 24 h post-injection. Interestingly, although the serum levels of both Fc- and Ig- fusions were similar, CSF and brain levels were slightly higher in hFc-IGF1R5 versus IgG-IGF1R5 (6.52 ± 2.00 vs. 5.11 ± 0.11; 11.27 ± 3.7 vs. 5.36 ± 0.58, respectively), suggesting that the molecular weight of the cargo may have an impact on its BBB permeability, although other causes, such as steric hindrance, should not be discarded. To confirm this observation, we generated multiple IGF1R5 constructs with molecular weights ranging from 80 kDa to 300 kDa and measured their transport across the in vitro BBB model (Appendix A). An inverse correlation was observed between Papp values and the MW of different IGF1R5 constructs, further supporting our in vivo observations. No change in BBB transport was observed for the A20.1 constructs with different sizes.

### 3.3. Hypothermic Properties of IGF1R5-hFc-Neurotensin in Rats and Mice

Next, we determined whether IGF1R5 is capable of delivering a pharmacologically active payload into the brain. Neurotensin and its receptors are highly expressed in the brain [30]. It was shown that neurotensin elicits hypothermia in rodents through neurotensin receptor type 1 (NTR1) located within the central nervous system [31]. Here, we first demonstrated that fusion constructs of IGF1R5-hFc-neurotensin were able to activate NTR1 in a cell-based assay (Figure 4A). Although a slight shift to the right was observed in the concentration–response curve when compared with the 13 amino-acid neuropeptide alone, this difference was not significant.

Intravenous injection of IGF1R5-hFc-neurotensin in rats and mice induced a dose-dependent drop in core body temperature (Figure 4B,F, respectively). We also observed that no change in core body temperature was observed when animals were injected with A20.1-hFc-neurotensin (Appendix A), which is consistent with its inability to cross the BBB. To further characterize the hypothermic effect of the IGF1R5-hFc-neurotensin constructs, three parameters were extrapolated from the dose–response curves; maximum response (Figure 4C,G), duration of response (Figure 4D,H) and area under the curve (Figure 4E,I). Quantification and statistical analysis of these parameters further demonstrated the hypothermic properties of IGF1R5-hFc-neurotensin, thus demonstrating that not only IGF1R5 constructs are able to cross the BBB but they are also able to deliver a pharmacologically active payload to target neurons.

### 3.4. Analgesic Properties of IGF1R5-mFc-Galanin and mFc-IGF1R5-Galanin in Rats

We further tested the ability of IGF1R5 to deliver a pharmacologically active payload by chemically conjugating galanin, which is a 29-amino-acid peptide that also does not cross the BBB, to its Fc fusion variants. Galanin was shown to reduce pain when directly injected into the brain [32]. We also demonstrated the analgesic properties of galanin when conjugated to single-domain antibodies that cross the BBB by RMT using the Hargreaves model of hyperalgesia [11,29]. In this model, paw inflammation was induced via subcutaneous injection of CFA in the hindpaw and reversal of hyperalgesia was observed via an increase in the latency of paw withdrawal from a thermal stimulus. Here, we demonstrated a dose–dependent analgesic response of galanin when conjugated to both mFc-IGF1R5 and IGF1R5-mFc (Supplemenatry Appendix A). Interestingly, when the reversal of hyperalgesia of mFc-IGF1R5-galanin and IGF1R5-mFc-galanin were compared at equimolar doses (4.96 mg/kg and 5.00 mg/kg, respectively), we noted that the analgesic effect was slightly higher in mFc-IGF1R5-galanin when compared with IGF1R5-mFc-galanin. This result is in accordance with what was observed in the BBB permeability assay in vitro, where the Papp values of IGF1R5 with mFc fusion in its C- or N-terminus were compared (Figure 3B).

## 4. Discussion

Despite growing knowledge and understanding of the pathophysiology of neurodegenerative and other brain diseases, the development of CNS drug candidates is severely hampered by poor tissue distribution due to the blood–brain barrier (BBB).

The BBB is responsible for protecting the brain from exposure to circulating toxins and pathogens. However, this protective function also presents a key challenge to the development of drugs targeting the CNS, in particular, biologics [1]. It was proposed that drug candidates could be delivered across the BBB via a process known as receptor-mediated transcytosis (RMT) [2], whereby ligands to receptors that transport large molecules across the BBB to supply the brain with nutrients needed for physiological homeostasis, such as transferrin receptor (TfR) or insulin receptor (IR), could be used to ‘shuttle’ attached therapeutic cargo molecules across the BBB. We previously generated single-domain antibodies (sdAbs) against different targets present in brain endothelial cells (BECs) that undergo RMT, including TMEM30A and IGF1R, and demonstrated their ability to cross the BBB in both in vitro and in vivo models [5,11,12,13].

In the present study, we demonstrated that IGF1R5, which is a camelid single-domain antibody (V_H_H) that targets IGF1R, transmigrated across the BBB and delivered diverse pharmacologically active payloads into the brain. IGF1R5 was humanized and may be considered a strong candidate for development as a BBB-delivery platform due to its stability, pH sensitivity and tolerance of different fusion formats.

One of the main advantages of targeting IGF1R for brain delivery of therapeutics is its enrichment in brain endothelial cells/brain microvessels when compared with peripheral tissue [33]. Furthermore, we recently showed that IGF1R transcript levels were twofold more abundant than another known RMT target, namely, TfR, in mouse BBB [33]. In our recent study [11], we demonstrated BBB crossing of a panel of IGF1R V_H_Hs in mice using an in situ brain perfusion technique, confirming that this BBB receptor could facilitate RMT of binding ligands, including antibodies.

Camelid V_H_Hs possess numerous desirable properties as BBB carriers, including high thermostability, resistance to proteases [34] and ease of optimization and engineering into various protein fusion formats. Llama V_H_H IGF1R5 demonstrated a high-affinity (sub-nanomolar) binding to IGF1R with a broad species cross-reactivity against human, rhesus, mouse and rat receptors. The linear epitope in the IGF1R structure recognized by the IGF1R5 that triggers structural re-arrangement of the receptor (and subsequent transcytosis) was mapped and shown to involve a single α-CT helix without activating any downstream signaling events [24].

Antibody humanization is a critical approach to eliminating or reducing the immunogenicity and improving the clinical translation of camelid antibodies; in contrast to shark-derived sdAbs (VNARs), which are difficult to humanize, humanization of camelid V_H_Hs has been routinely achieved. The main challenge in the humanization process is to maintain the full biological function while reducing the risk of adverse side effects [35]. Through CDR grafting and resurfacing methods, we successfully generated a series of humanized variants of IGF1R5. One of these variants, namely, IGF1R5-H2, acquired a target-binding profile that was characterized by slightly weaker binding affinities and accelerated ‘off-rates’ compared with the parent camelid variant, which is a feature that is considered advantageous for RMT carriers where fast endosomal dissociation from the target receptor may facilitate its abluminal release. This humanized variant also displayed weaker binding affinity at acidic pH, which is another desirable characteristic previously demonstrated to facilitate the abluminal release of TfR-binding antibodies [36]. In our subsequent studies, the IGF1R5-H2 sdAb variant demonstrated improved transcytosis compared to the parent IGF1R5 sdAb in the rat BBB model in vitro, while in the human BBB model, both variants had similar Papp values. Further studies employing the pulse–chase method or dynamic analyses of trafficking through intracellular endosomal compartments of these two variants will be necessary to dissect the potential benefits to transcytosis efficiency imparted by different levels of affinity and pH sensitivity of their binding to the target receptor.

Due to their size (~15 kD), sdAbs have a short plasma half-life and are rapidly cleared from the circulation by glomerular filtration in the kidney, thus limiting their use as systemic therapeutics. Numerous strategies are available to improve the pharmacokinetic profiles of sdAbs, including their fusion to the Fc region of IgG. The tolerance of BBB carrier sdAbs to fusion at both N- and C-terminus is advantageous, as it creates a platform that can be used for different types of therapeutic payloads. For example, for monoclonal antibodies, it is preferable to fuse BBB carriers to their C-terminus, as it would not affect the antibody target binding. Our data indicated that IGF1R5 BBB permeability was not affected in fusions through either its N- or C-terminus.

We next examined the serum, CSF and brain levels of hFc-IGF1R5 24 h after its systemic administrations. As expected, a significant amount of hFc-IGF1R5 was still present in the serum samples, suggesting that the hFc fusion greatly increased the half-life of IGF1R5. This was not surprising since, unlike high-affinity TfR antibodies that were shown to have accelerated systemic clearance due to peripheral target-mediated clearance [37], we showed that IGF1R constructs have pharmacokinetic profiles that are comparable to control monoclonal antibodies [23]. hFc-IGF1R5 levels were significantly increased in both CSF and capillary-depleted brain fractions. Similar to Fc fusions, IgG-IGF1R5 bi-specific antibodies demonstrated considerable accumulation in CSF and brain tissue. Interestingly, we noted that levels in brain fractions were slightly reduced for IgG fusions when compared with the Fc format, suggesting that the size of the molecule may have an impact on BBB permeability.

For drug development purposes, it is necessary to demonstrate that sdAbs are capable of transporting a pharmacologically active payload into the CNS following RMT in preclinical models using rodents and non-human primates. SPR data acquired in our study demonstrated the binding cross-reactivity of IGF1R5 across different species, providing great versatility in the choice of preclinical models. This is an important departure from the preclinical studies involving TfR that required the use of transgenic animals expressing human TfR since those antibodies showed no species cross-reactivity [38]. We fused the neuropeptide neurotensin to the C-terminus of IGF1R5-hFc and measured its pharmacological effect. Neurotensin (NT) is a 13-amino-acid neuropeptide that is widely distributed in the CNS that mediates its effects, mainly through neurotensin receptor type 1 (NTR1), a G-protein-coupled receptor [31]. It was shown that central injection of NT leads to a sustained decrease in core body temperature [39]. This effect was also determined to be mediated by NTR1 since, in NTR1 knockout mice, NT administration failed to induce changes in body temperature [40]. Additionally, analogs that are selective for other subtypes of NT receptors did not induce hypothermia, further supporting the involvement of the NTR1 receptor subtype in mediating this effect [41]. In the present study, we demonstrated that IGF1R5-hFc-neurotensin constructs induced a dose-dependent hypothermic response in both mice and rats. Interestingly, although the levels of IGF1R5 constructs are elevated even 24 h post-injection, the effects of IGF1R5-hFc-neurotensin lasted a maximum of 290 and 95 min in mice and rats, respectively, at the highest dose studied. We speculate that feedback mechanisms, including thyroid activity and brown adipose tissue stimulation, play a role in normalizing the core temperature to baseline levels.

We also measured the analgesic properties of galanin when conjugated to IGF1R5. Galanin is a neuropeptide of 29 amino acids (30 in humans) that was originally isolated from a porcine intestine [42]. Galanin effects are mediated by three subtypes of galanin receptors (Gal R1, Gal R2 and Gal R3) that belong to the family of G-protein coupled receptors [43]. The analgesic properties of galanin were further demonstrated by studies showing that direct injection of galanin into the central nervous system has antinociceptive effects in different experimental models of pain [32,44]. Here, we showed that IGF1R5 sdAb Fc fusion constructs conjugated with galanin were capable of reversing hyperalgesia of rats in the Hargreaves model. Interestingly, in accordance with what we observed in the BBB model in vitro, mFc-IGF1R5-galanin constructs performed slightly better than IGF1R5-mFc-galanin. Different efficacy of these constructs could be the result of attenuation/modification of either IGF1R5 or galanin functionality or both.

The hypothermic and analgesic effects of neurotensin and galanin, respectively, when conjugated to IGF1R5 further confirmed the delivery of pharmacologically active amounts of payloads to the site of actions in the CNS. Taken together, we demonstrated the feasibility of delivering cargos into the brain with single-domain antibodies against IGF1R, notably IGF1R5, which permeated the BBB through RMT, highlighting its potential use for the development of drugs targeting the central nervous system.

## 5. Conclusions

In conclusion, we demonstrated that IGF1R5, which is a sdAb targeting IGF1R in the BBB, is a promising carrier for delivering drugs targeting the central nervous system. This antibody can be presented in different formats and fused to payloads of variable sizes, further highlighting its clinical potential, especially considering that poor brain distribution is one of the major limitations of drugs developed for CNS applications.

## Figures and Tables

**Figure 1 pharmaceutics-14-01452-f001:**
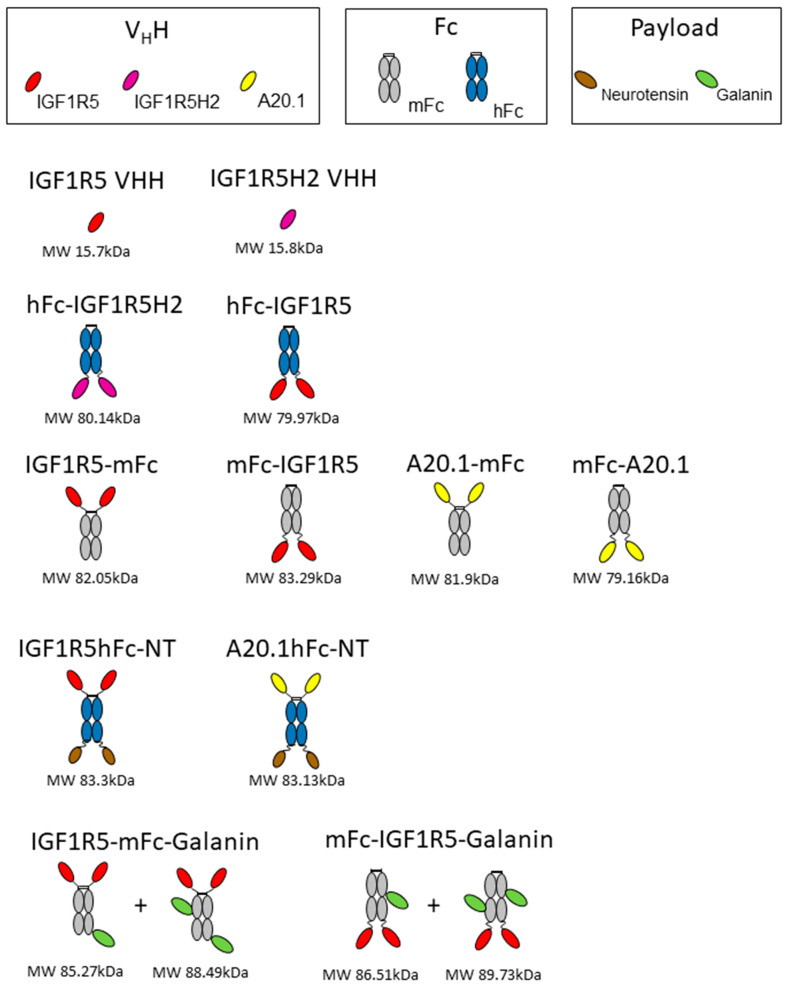
A schematic figure depicting the different constructs used in this study. Three different V_H_Hs, Fc regions with human and mouse origins, and two neuropeptides (neurotensin and galanin) were used to generate the antibodies. The predicted molecular weight (MW) for each construct is also shown.

**Figure 2 pharmaceutics-14-01452-f002:**
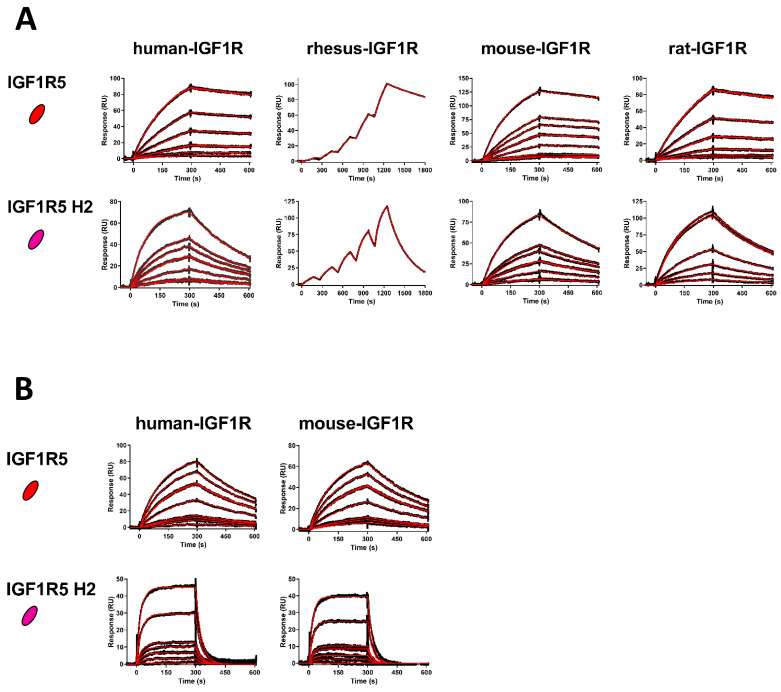
High-affinity binding of IGF1R5 V_H_Hs to IGF1R ectodomains. (**A**) SPR sensorgrams demonstrating wild-type IGF1R5 and humanized IGF1R5-H2 V_H_Hs binding to surface-immobilized human, rhesus, mouse and rat IGF1R (pH 7.4, 25 °C). V_H_H concentrations in flow ranged from 0.25 to 10 nM (IGF1R5) and from 1 to 25 nM (IGF1R5-H2). Kinetics and affinities were determined using multi-cycle kinetics (human, mouse, rat IGF1R) or single-cycle kinetics (rhesus IGF1R) analyses. (**B**) Sensorgrams demonstrating the binding of V_H_Hs to human and mouse IGF1R at acidic pH (pH 5.6, 37 °C). V_H_H concentrations in flow ranged from 0.25 to 10 nM (IGF1R5) and from 1 to 50 nM (IGF1R5-H2). Black lines: raw data; red lines: 1:1 binding model fitting.

**Figure 3 pharmaceutics-14-01452-f003:**
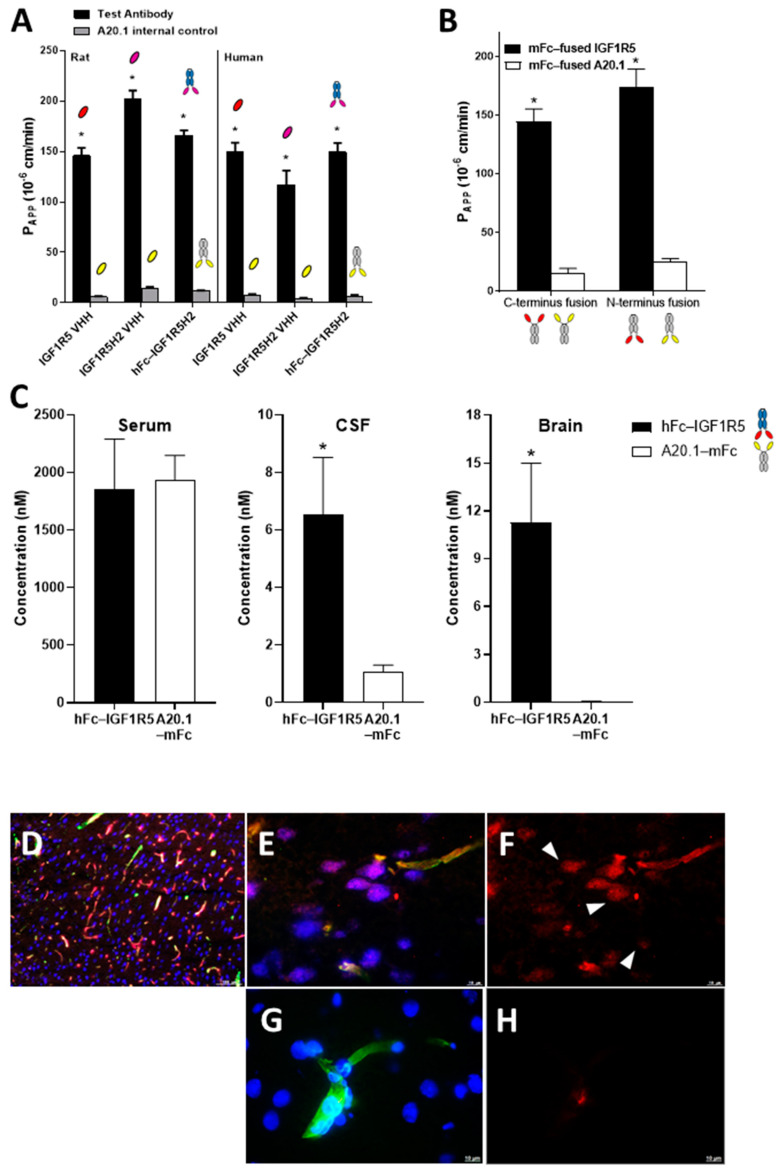
**In vitro and in vivo BBB transport of IGF1R5 in the V_H_H format or in hFc or mFc fusions.** (**A**) Transport of IGF1R5 in the V_H_H format, and humanized (H2) and hFc fused variants of IGF1R5 across a rat (SV-ARBEC) and human (iBEC) in vitro BBB model. Antibodies were applied in the upper compartment of the BBB insert and then quantified over time in the bottom chamber using SRM to determine Papp values. Papp values (cm/min) of antibodies are shown as means ± sd derived from 6 separate transwell inserts. * *p* < 0.05 vs. respective negative control. (**B**) Transport of mFc-IGF1R5 and IGF1R5-mFc across a rat BBB model in vitro. Antibodies were paired with corresponding controls in the upper compartment of the BBB insert and then quantified over time in the bottom chamber using SRM to determine the Papp values. Papp values (cm/min) of antibodies are shown as means ± sd derived from 6 separate transwell inserts. * *p* < 0.05 vs. respective negative control. (**C**) Concentrations of hFc-IGF1R5 or A20.1-mFc in serum, CSF or capillary-depleted brain of rats at 24 h following a bolus i.v. injection of 15 mg/kg of each antibody. The concentrations were measured using SRM analysis in at least 3 animals and the bars represent mean and SD. * *p* < 0.01 vs. A20.1-mFc. (**D**–**G**) Immunofluorescence staining of the rat frontal cortex 48 h after intravenous injection of 15 mg/kg of IGF1R5-mFc (**D**–**F**) or an equimolar dose of A20.1-mFc (**G**,**H**). Brain vessels were detected using RCA-1 (green); neurons were detected using antibodies against NeuN (blue); IGF1R5-mFc (red). Images were observed at 10x objective (**D**) and 60x (**E**–**H**). Scale bars, 100 µm and 10 µm.

**Figure 4 pharmaceutics-14-01452-f004:**
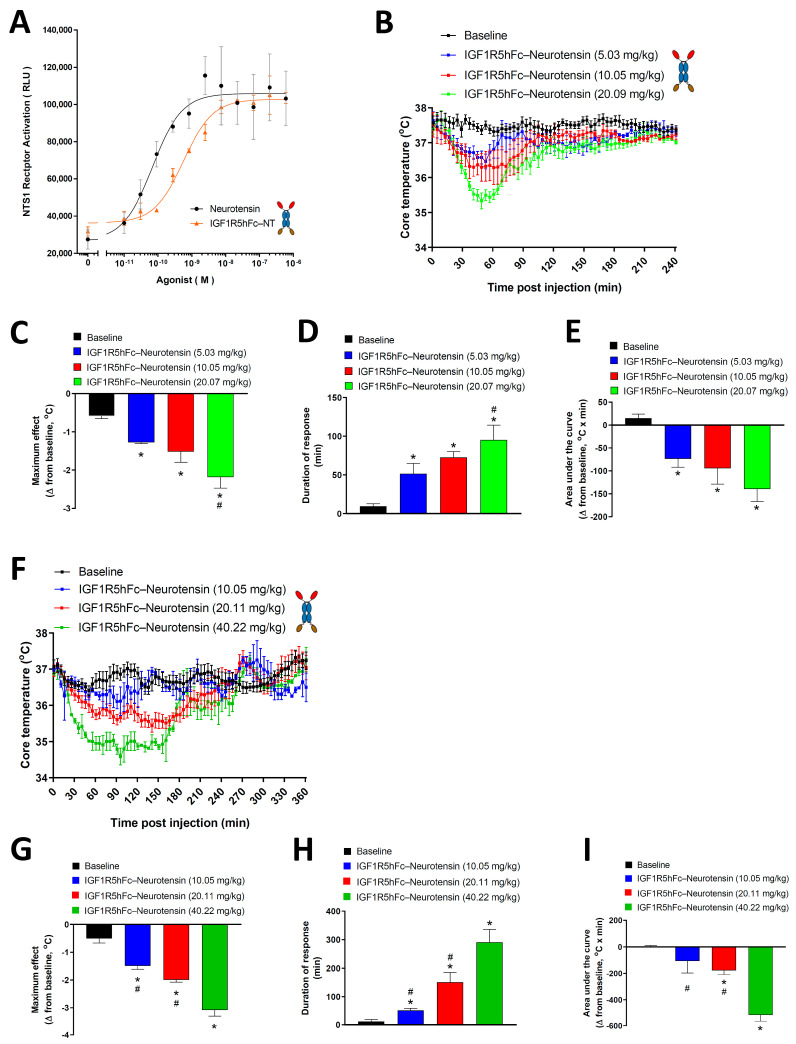
**NTR1 activation and IGF1R5-hFc-neurotensin-induced hypothermia in rats and mice.** (**A**) Concentration–response curve of neurotensin and IGF1R5-hFc-NT-induced activation of NTR1 in a cell-based assay. (**B**) Dose–response curve of the hypothermic effects of IGF1R5-hFc-neurotensin in rats (5.03 mg/kg, *n* = 3; 10.05 mg/kg, *n* = 6; 20.07 mg/kg, *n* = 4). Core body temperature was monitored using telemetry up to 4 h post intravenous injection of test compounds and the maximum response (**C**), duration of response (**D**) and area under the curve (**E**) were obtained. (**F**) Dose–response curve of the hypothermic effects of IGF1R5-hFc-neurotensin in mice (10.05 mg/kg, *n* = 2; 20.11 mg/kg, *n* = 7, 40.22 mg/kg, *n* = 2). Core body temperature was monitored using telemetry up to 6 h post-intravenous injection of test compounds and maximum response (**G**), duration of response (**H**) and area under the curve (**I**) were obtained. Results are mean ± SEM of 2–8 animals in each group. * *p* < 0.05 vs. baseline; # *p* < 0.05 vs. highest dose injected (20.06 mg/kg in rats and 40.22 mg/kg in mice).

**Table 1 pharmaceutics-14-01452-t001:** SPR-derived kinetics and equilibrium dissociation constants for V_H_H-IGF1R interactions. ^1^ Determined at pH 7.4, 25 °C; ^2^ determined at pH 5.6, 37 °C.

IGF1R Ectodomain	pH	IGF1R5			IGF1R5-H2		
		*k*_a_ (M^−1^s^−1^)	*k*_d_ (s^−1^)	*K*_D_ (M)	*k*_a_ (M^−1^s^−1^)	*k*_d_ (s^−1^)	*K*_D_ (M)
Human ^1^	7.4	5.3 × 10^5^	3.4 × 10^−4^	6.4 × 10^−10^	3.8 × 10^5^	2.9 × 10^−3^	7.6 × 10^−9^
Rhesus ^1^	7.4	1.5 × 10^6^	5.9 × 10^−4^	4.0 × 10^−10^	2.3 × 10^5^	3.8 × 10^−3^	1.7 × 10^−8^
Mouse ^1^	7.4	2.6 × 10^5^	3.0 × 10^−4^	1.1 × 10^−9^	2.3 × 10^5^	2.1 × 10^−3^	9.1 × 10^−9^
Rat ^1^	7.4	3.4 × 10^5^	3.8 × 10^−4^	1.1 × 10^−9^	2.3 × 10^5^	2.6 × 10^−3^	1.1 × 10^−8^
Human ^2^	5.6	6.9 × 10^5^	2.8 × 10^−3^	4.0 × 10^−9^	2.1 × 10^5^	3.4 × 10^−2^	1.6 × 10^−7^
Mouse ^2^	5.6	5.9 × 10^5^	2.8 × 10^−3^	4.8 × 10^−9^	2.9 × 10^5^	4.3 × 10^−2^	1.5 × 10^−7^

## Data Availability

Not applicable.

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
