# Peer review of "Brain Delivery of IGF1R5, a Single-Domain Antibody Targeting Insulin-like Growth Factor-1 Receptor"

_pharmaceutics, 2022, doi:10.3390/pharmaceutics14071452_

Round 1

Reviewer 1 Report

Brain delivery of IGF1R5, a single-domain antibody targeting insulin-like growth factor-1 receptor

Summary: The authors describe novel fusion proteins capable of crossing the blood brain barrier and delivering drug payloads. The introduction and methods section are well written and include sufficient detail and references. The results are supported by data presented. This is a high quality manuscript with great importance and novelty. However, some critical data is missing and I believe the manuscript can be reevaluated for publication when the following comments are addressed.

1)    Introduce IGF1R5-H2 in results section: humanization was described only in methods section. Including few lines about IGF1R5-H2 in results as well.

2)    Figure 2A: include data for negative control and also perform appropriate statistics

3)    Figure 2D: While a secondary control (included) is relevant, an important negative control would be A20.1.Fc. Include this as well. Figure 2D is also missing labeling. Is the Fc staining in parenchyma significantly increased over control included? Define ROIs across the image, quantify channel intensity and perform statistical analysis to compare before making any conclusions 

4)    Lines 400 and 413: Should it be” intravenously injected anti-Fc-IGF1R5 antibody” or  ” intravenously injected Fc-IGF1R5” 

5)    Fc domain fused to IGF1R5 is mouse or human? Clarify and maintain consistency. There are discrepancies in the manuscript 

6)    Consider including a table with all the constructs (fusion proteins) referred to in the manuscript, including their expected molecular weights and any characterization information (purity and MW by SDS-PAGE at the minimum) 

7)    Provide complete sequences and plasmid maps for all the constructs in supplementary section

8)    In figure 2B, Fc-IGF1R5 and IGF1R5-Fc are compared. Throughout rest of the manuscript, Fc fused IGF1R refers to which orientation?

9)    Figure 2A: Is the experimental group hFc-IGF1R5-H2 or hFc-IGF1R5? There is discrepancy between figure axis and in-text/figure legend. 

10) The manuscript  would greatly benefit from schematics depicting the fusion proteins under evaluation

11) Maintain consistency in rereferring to molecules throughout text, figures and supplementary section. Improved labeling is necessary. For example, IGF1R5 is also referred to as IGF1R5 VHH in some places. Fc-IGF1R5 is also called hFc-IGF1R5, Fc fused IGF1R5 and Fc fused IR5. A20.1-mFc is referred to as A20.1Fc at some places 

12) Show basic characterization for purified proteins (at least SDS-PAGE for purity assessment)

13) Neurotensin and Galanin fusion protein architecture is not well described, no characterization either (how many galanin/molecule), how are they fused?

14) Line 478: equimolar is in reference to galanin or entire fusion protein?

15) Explain why IGF1R5-H2 was not evaluated extensively

Author Response

We would like to thank the reviewer for the time and review of our manuscript. We have taken the comments into account and revised our manuscript accordingly. Detailed replies to the comments are listed below.

  1. Introduce IGF1R5-H2 in results section: humanization was described only in methods section. Including few lines about IGF1R5-H2 in results as well.

In agreement with the reviewer comment we have modified the text of the revised manuscript and IGF1R5-H2 has been introduced in the Results section 3.1.

  1. Figure 2A: include data for negative control and also perform appropriate statistics

We would like to than the reviewer for this comment. In each well, the appropriate format of the negative control (A20.1) was used and their Papp values determined. In this version of the manuscript, the negative control data and statistical analyses have been added to Fig 2A.

  1. Figure 2D: While a secondary control (included) is relevant, an important negative control would be A20.1.Fc. Include this as well. Figure 2D is also missing labeling. Is the Fc staining in parenchyma significantly increased over control included? Define ROIs across the image, quantify channel intensity and perform statistical analysis to compare before making any conclusions 

We have modified Fig 2D to provide A20.1-Fc and images at higher magnification.  We did not perform quantification of immunofluorescence in sufficient number of brains for statistical analyses, mostly because we focussed on (absolute) quantitative measurements of brain levels of injected molecules (MS analyses that allow quantification of various antibody/fusion protein domains as well as their definitive sequence-based identification in the rain and body fluids – something that cannot be achieved by immunofluorescence); therefore these images are meant as a visual demonstration not quantitative assessment.  

  1. Lines 400 and 413: Should it be” intravenously injected anti-Fc-IGF1R5 antibody” or  ” intravenously injected Fc-IGF1R5” 

We would like to thank this reviewer for point out this error. It should have read “intravenously injected Fc-IGF1R5”. We have made modifications to the revised version of the manuscript and the proper description of the methodology has been used.

  1. Fc domain fused to IGF1R5 is mouse or human? Clarify and maintain consistency. There are discrepancies in the manuscript 

We apologize for the inconsistences found in the manuscript. The text has been thoroughly reviewed to maintain consistency of nomenclature. Use of both mouse and human Fc in different experiments has been clarified throughout.

  1. Consider including a table with all the constructs (fusion proteins) referred to in the manuscript, including their expected molecular weights and any characterization information (purity and MW by SDS-PAGE at the minimum)

We would like to thank the reviewer for this comment. We have modified the manuscript and Figure 1 now includes the schematics of all molecules used in this study and their molecular weights.  We have also included further characterization of some of these molecules is in the Supplementary material. Supplementary Figure S2 contains representative SDS-PAGE and a representative silver stained image following the chemical conjugation of galanin.

  1. Provide complete sequences and plasmid maps for all the constructs in supplementary section

We would like to thank the reviewer for this comment. The sequences of the constructs are present in reference 24 of the manuscript. In addition to the references, a descriptions has also been included to the Methods Section 2.9 of the revised manuscript.

  1. In figure 2B, Fc-IGF1R5 and IGF1R5-Fc are compared. Throughout rest of the manuscript, Fc fused IGF1R refers to which orientation?

We apologize for the confusion in the orientation of the Fc fusion throughout the manuscript. In this submitted version, the text has been thoroughly reviewed and revised to maintain consistency of nomenclature. Use of mouse or human Fc fusions with specific orientation has been specified for each experiment.

  1. Figure 2A: Is the experimental group hFc-IGF1R5-H2 or hFc-IGF1R5? There is discrepancy between figure axis and in-text/figure legend. 

We apologize for the discrepancy. In this figure, the experimental group is hFc-IGF1R5-H2. The inconsistency has been corrected in the revised version of the manuscript.

  1. The manuscript would greatly benefit from schematics depicting the fusion proteins under evaluation

We would like to thanks the reviewer for this comment. Following the suggestion of this reviewer, a schematic of various antibody formats used in the manuscript has been added as Figure 1, as well as in other figures for ease of interpretation.

  1. Maintain consistency in rereferring to molecules throughout text, figures and supplementary section. Improved labeling is necessary. For example, IGF1R5 is also referred to as IGF1R5 VHH in some places. Fc-IGF1R5 is also called hFc-IGF1R5, Fc fused IGF1R5 and Fc fused IR5. A20.1-mFc is referred to as A20.1Fc at some places 

We apologize for the inconsistency present in the manuscript. In this revised version, the text has been thoroughly edited for consistency of nomenclature.

  1. Show basic characterization for purified proteins (at least SDS-PAGE for purity assessment)

Following the suggestion of the reviewer, the characterization (SDS-PAGE) for different molecules has been included in Supplementary Figure S2

  1. Neurotensin and Galanin fusion protein architecture is not well described, no characterization either (how many galanin/molecule), how are they fused?

We would like to thank the reviewer for tis comment. In this version of the manuscript we have provided additional description to clarify the process of obtaining the neurotensin and galanin constructs in the methods section, as well as in Supplementary Figure S2. Briefly, for neurotensin we have generated plasmids that encode a linker and the sequence for neurotensin at the C-terminus of the human Fc sequence used. This plasmid was then transfected to CHO-3E7 for protein production. Thus, each fusion molecule of IGF1R5hFc-Neurotensin contains two molecules of neurotensin. A SDS-PAGE image of the purified constructs is present in Supplemental figure 2. Galanin on the other hand was chemically conjugated using cysteamide modified galanin and maleimide-activated IGF1R5 constructs. We titrated the reaction to achieve about 1 to 2 galanin molecules per construct. Silver stained SDS-PAGE was used to confirm a shift in molecular weight size after conjugation. A representative image in present in Supplementary Figure S2 of the revised manuscript.

  1. Line 478: equimolar is in reference to galanin or entire fusion protein?

We apologize for the ambiguity. The equimolar adjustments were made to full fusion proteins and the exact doses are now specified in the revised manuscript.

  1. Explain why IGF1R5-H2 was not evaluated extensively

IGF1R5-H2 has been evaluated in therapeutic bi-specific program(s) which are under embargo for disclosures. IGF1R5-H2 retains most characteristics and demonstrates some improvements compared to IGF1R5 as shown in this study using BBB model in vitro.

Reviewer 2 Report

1. What is the difference between Fc-IGF1R5 and conventional antibody with two heavy chain and two light chain?

2. Can the conventional antibody overcome the question of short serum half-life of sdAbs? The author should provide data about the half-life of IGF1R5, Fc-IGF1R5 and conventional anti-IGF1R.

3. What is the advantage of Fc-IGF1R5 over conventional antibody? After Fc conjugation, the non-specific interactions of Fc-IGF1R5 with tissues expressing high levels of Fc receptors may be similar to conventional antibody, which may delete the small-size advantage of sdAbs.

4. Table 1. The unit of Ka should be M-1s-1, while the unit of Kd should be s-1.

5. Figure 2A. Statistics should be added.

6. Figure 2D. Almost all staining was shown inside the vessels. The staining outside the vessels should be labeled with arrows to demonstrate the successful BBB crossing. It is better if the authors can also label the vessels using markers such as CD31.

7. Supplemental Figure S2B, the BBB penetration between IGR1R5 constructs with MW 300kDa and A20.1 was almost not significant, meaning no BBB crossing. So if the IGR1R5 can be used to modify nanoparticles to mediate brain-targeted drug delivery?

8. Supplemental Figure S4. Statistics should be added.

9. Some important articles on BBB crossing for brain-targeting drug delivery should be added. For example, Mfsd2a inhibition (Adv Healthc Mater 2021, 10, 2001997), LRP1 upregulation (J Control Release 2019, 303, 117), and use of gp96 (Adv Sci (Weinh), 2022, 9(16): e2105854).

Author Response

We would like to thank the reviewer for the time and review of our manuscript. We appreciate the helpful comments and suggestions of this reviewer. Our manuscript has been carefully revised according to the comments and suggestions. Detailed replies to the comments are listed below.

  1. What is the difference between Fc-IGF1R5 and conventional antibody with two heavy chain and two light chain?

Note that IGF1R5 fusion with Fc can be accomplished on either its N- terminus (IGF1R5-Fc; ‘looks’ similar to IgG) or on C-terminus (Fc-IGF1R5 – applicable as platform for easy generation of bi-specific antibodies).  In both cases, these fusion proteins are MW-80kD compared to MW-150KD of full IgG.

  1. Can the conventional antibody overcome the question of short serum half-life of sdAbs? The author should provide data about the half-life of IGF1R5, Fc-IGF1R5 and conventional anti-IGF1R.

Single domain antibody IGF1R5 is a BBB carrier that has modularity of fusion to any other protein cargo (in other words, IGF1R5 VHHs are not intended to be used ‘on their own’ but attached to therapeutic molecules).  Therefore, IGF1R5 VHH will mostly ‘assume’ half-life of the protein cargo it is attached to, since its own binding to peripheral target is limited – if fused to a therapeutic antibody (that has no peripheral target), the bi-specific antibody will have a very long serum half-life beneficial for extended brain exposure. If fused to a non-antibody protein with MW higher than renal clearance, it will largely ‘assume’ the half-life of that protein.  The advantage of sdAb format for this application is its modularity and small size (much easier to engineer, humanize, and incorporate into fusion proteins compared to a full IgG as ‘BBB carrier’).

  1. What is the advantage of Fc-IGF1R5 over conventional antibody? After Fc conjugation, the non-specific interactions of Fc-IGF1R5 with tissues expressing high levels of Fc receptors may be similar to conventional antibody, which may delete the small-size advantage of sdAbs.

Fc fusions with IGF1R5 used in this study are ‘model molecules’ to demonstrate a versatility of fusion formats and to assess brain penetration/exposure when the serum half-life of the BBB carrier is extended via FcRn-mediated recycling (the same mechanism by which full monoclonal antibodies achieve long serum half-life).  FcRn receptor is expressed by endothelial cells and will drive internalization and recycling of Fc-containing molecules, making their serum half-life similar to that of full monoclonal antibodies,

  1. Table 1. The unit of Kashould be M-1s-1, while the unit of Kdshould be s-1. 

We would like to thank the reviewer for pointing the error in the units. This has been corrected in the revised version of the manuscript.

  1. Figure 2A. Statistics should be added.

Following the comments of this reviewer, the statistical analyses have been added to Figure 2A.

  1. Figure 2D. Almost all staining was shown inside the vessels. The staining outside the vessels should be labeled with arrows to demonstrate the successful BBB crossing. It is better if the authors can also label the vessels using markers such as CD31.

In this revised version of the manuscript we have modified this figure (now reads figure 3). Arrows and enlarged images indicating parenchymal staining are added to the Figure 3F.  

  1. Supplemental Figure S2B, the BBB penetration between IGR1R5 constructs with MW 300kDa and A20.1 was almost not significant, meaning no BBB crossing. So if the IGR1R5 can be used to modify nanoparticles to mediate brain-targeted drug delivery?

All molecules shown in Figure S3B (originally Figure S2B) are fusion proteins with IGF1R5 with various degree of complexity. One biophysical parameter that can restrict the BBB crossing of these fusion proteins is their molecular weight (hydrodynamic space). The dimensions of endocytic vesicles (clathrin-coated pits – 100 nm; caveolae – 50 nm) are the limiting parameter for RMT route of transport, particularly for nanoparticles; these dimensions are permissive for large protein molecules, including antibodies (6-15 nm).  In addition to size, other issues could cause lower BBB crossing of very large fusion proteins (300 kD; as shown in Figure S2B)  – such as steric hindrance of fused IGF1R5 VHH (15 kD) within a large fusion protein, preventing its optimal interaction with the target receptor.   

Nanocarriers can be targeted with IGF1R5 for enhanced BBB delivery (provided the overall dimensions of these construct are significantly less than 100 nm) – however, this is not the subject of this manuscript and we opted not to demonstrate this further.

  1. Supplemental Figure S4. Statistics should be added.

In accordance to this reviewer comment, statistical analysis have been included in Figure S5 (originally Figure S4)

  1. Some important articles on BBB crossing for brain-targeting drug delivery should be added. For example, Mfsd2a inhibition (Adv Healthc Mater 2021, 10, 2001997), LRP1 upregulation (J Control Release 2019, 303, 117), and use of gp96 (Adv Sci (Weinh), 2022, 9(16): e2105854).

We would like to thank the reviewer for this comment. We have included some additional references on RMT receptors in the Introduction section of the revised manuscript.

Important to note is that a) transport through Mfsd2a pathway is not by RMT mechanism, rather via a regulation of bulk transcytosis; b) there is recent evidence (and much debate) about lack of LRP1 expression in brain endothelial cells – therefor blood-to-brain LRP1-mediated RMT across BBB remains debatable;  c) delivery of cargo across the BBB using variously engineered and targeted nanocarriers is another vast (and controversial) field with an extensive published work – since the focus of this manuscript is not on nanocarriers, we felt that singling out one publication/technology among a vast body of published work would not be appropriate without detailed comparative analyses.

Reviewer 3 Report

1.     Please provide biological experimental data, such as Wester Blot to  prove IGF1R5 against IGF1R are suitable as RMT carriers for delivery of therapeutic cargoes for CNS applications.

2.     Please state the number of animals in the animal experiment (n number). 

3.     After binding the IGF1R5, how can authors prove that it does not affect other biological functions?? please discuss about the possible side effect and off target outcome?

4.     Please provide the experiment results on:  after the IGF1R5 is knocked out, what is the situation and efficiency on  BBB crossing ???

Author Response

We would like to thank the reviewer for the time and review of our manuscript. Detailed replies to the comments are listed below.

  1. Please provide biological experimental data, such as Wester Blot to prove IGF1R5 against IGF1R are suitable as RMT carriers for delivery of therapeutic cargoes for CNS applications.

In the present study we have provided biological data demonstrating suitability of IGF1R5 as BBB carrier of various payloads, including absolute quantification of delivered molecules in body fluids and bran tissues, as well as pharmacodynamics data with neurotensin and galanin as payloads. Although we are not sure as to what WB the referee is asking for we would like to mention that mass spectrometry data provided is more accurate, sensitive and quantitative than any WB technique.

  1. Please state the number of animals in the animal experiment (n number).

Following this comment in the revised manuscript we have added the number of animals for each animal experiment in the respective figure legend.

  1. After binding the IGF1R5, how can authors prove that it does not affect other biological functions?? please discuss about the possible side effect and off target outcome?

In our recent publication (Sheff et al. 2021, reference 25 in the manuscript) we have demonstrated that IGF1R5 binding to IGF1R does not trigger signaling through the receptor, therefore not interfering with the physiological function of the receptor.  It is also worth noting that anti-IGF1R monoclonal antibodies developed for the treatment of cancer, with a mechanism of action of actively inhibiting the signaling through the receptor and its removal from the membrane did not trigger any significant side effects in clinical trials. Although hyperglycemia is the most common side effects of anti-IGF-1R mAbs therapy, it is reported to be mostly of grades 1–2, which can be controlled with oral diabetic medications with continued monoclonal antibody treatment. Since the exact mechanism for this side effect is currently unknown, modeling and predicting this potential side effect with the use of IGF1R5 therapy is not possible and should be the subject of well-defined toxicology studies, which is not in the scope of the current manuscript.

  1. Please provide the experiment results on: after the IGF1R5 is knocked out, what is the situation and efficiency on  BBB crossing ???

We have not perform IGF1R (we assume this is what reviewer meant) knockout to address this question. IGF1R knockout is lethal, and the only possibility to explore the effect of lowering IGF1R in BBB on transport is in heterozygous or conditional knockout animals. However, these studies may be redundant in view of an unequivocal confirmation of IGF1R5 crossing the BBB provided in this study.

Round 2

Reviewer 1 Report

Most of the comments have been addressed. I believe the manuscript is suitable for publication.

Reviewer 2 Report

The authors have addressed all my concerns.